# Structure of the scaffolding protein and portal within the bacteriophage P22 procapsid provides insights into the self-assembly process

Hao Xiao[1,2☉], Wenyuan Chen[1☉], Hao Pang[1,3], Jing Zheng[1], Li Wang[2], Hao Feng[2], Jingdong Song[3*], Lingpeng Cheng[1*], Hongrong Liu[1*]

**1** Institute of Interdisciplinary Studies, Key Laboratory for Matter Microstructure and Function of Hunan Province, Key Laboratory of Low-dimensional Quantum Structures and Quantum Control, School of Physics and Electronics, Hunan Normal University, Changsha, China, **2** College of Life Sciences, Hunan Normal University, Changsha, China, **3** National Institute of Pathogen Biology, Chinese Academy of Medical Sciences & Peking Union Medical College, Beijing, China

☉ These authors contributed equally to this work.
* songjingdong@ipbcams.ac.cn (JS); lingpengcheng@hunnu.edu.cn (LC); hrliu@hunnu.edu.cn (HL)

## Abstract

In the assembly pathway of tailed double-stranded DNA (dsDNA) bacteriophages and herpesviruses, a procapsid with a dodecameric portal for DNA delivery at a unique vertex is initially formed. Appropriate procapsid assembly requires the transient presence of multiple copies of a scaffolding protein (SP), which is absent in the mature virion. However, how the SP contributes to dodecameric portal formation, facilitates portal and coat protein incorporation, and is subsequently released remains unclear because of a lack of structural information. Here, we present the structure of the SP–portal complex within the procapsid of bacteriophage P22 at 3–9 Å resolutions. The AlphaFold2-predicted SP model fits well with the density map of the complex. The SP forms trimers and tetramers that interact to yield a dome-like complex on the portal. Two SP domains mediate multimerization. Each trimer interacts with two neighboring portal subunits. The SP has a loop-hook-like structure that aids in coat protein recruitment during viral assembly. The loops of those SP subunits on the portal are positioned in clefts between adjacent portal subunits. Conformational changes in the portal during phage maturation may trigger the disassembly and release of the SP complex. Our findings provide insights into SP-assisted procapsid assembly in bacteriophage P22 and suggest that this strategy is also implemented by other dsDNA viruses, including herpesviruses.

## Introduction

Tailed double-stranded DNA (dsDNA) bacteriophages and their eukaryotic counterparts, such as herpesviruses, possess a dodecameric portal at a unique vertex of the capsid; this portal serves as a channel for DNA packaging during viral assembly and delivery [1–3]. During assembly, a precursor capsid (procapsid or prohead) is initially formed. The portal interacts with multiple copies of a scaffolding protein (SP) to form a nucleation complex, which

**Data availability statement:** The electron density maps, atomic coordinates, and the pseudo-atomic model have been deposited in the EM Data Bank and Protein Data Bank under accession code EMD-61452, EMD-61453, EMD-61454, EMD-61455, EMD-61456, EMD-61457, EMD-61460, EMD-61461, 9JGA, 9JG6, 9KYV, 9KYW, 9KYX, and 9KYY. The software package for cryo-electron microscopy of virus icosahedra, symmetry-mismatch, and local reconstruction has been deposited in Zenodo (https://doi.org/10.5281/zenodo.8378566).

**Funding:** This research was supported by the National Science and Technology Major Project of China (2023ZD0500501 to H.L.), the National Natural Science Foundation of China (12034006, 32430020 and 32071209 to H.L., 32371263 to L.C., 32401014 to H.X., 32200994 to W.C., 31971122 to L.C.), the Natural Science Foundation of Hunan Province, China (2023JJ30379 to L.C., 2024JJ6304 to H.X.), the science and technology innovation Program of Hunan Province (2024RC3150 to W.C.), and the Science Foundation for the State Key Laboratory for Infectious Disease Prevention and Control of China (2022SKLID203 to J.S.). The funders had no role in study design, data collection and analysis, decision to publish, or preparation of the manuscript.

**Competing interests:** The authors have declared that no competing interests exist.

**Abbreviations:** cryo-EM, cryo-electron microscopy; CsCl, cesium chloride; dsDNA, double-stranded DNA; PR, phase residual; SP, scaffolding protein; TFS, Thermo Fisher Scientific.

initiates the assembly of the major capsid protein (i.e., coat protein) and other minor proteins into a procapsid [4–7]. In phages such as HK97 and T5, the scaffold exists as a scaffolding domain covalently linked to the N-terminus of the coat protein [8,9]. An ATP-dependent terminase complex packs the viral dsDNA into the procapsid through the portal [7]. In parallel with the DNA package and viral maturation, the scaffold within the procapsid is either degraded (for phages lambda, T4, and HK97 and herpesviruses) or recycled for further rounds of assembly (for phages P22 and Phi29) [10]. Pressure from the packaged DNA induces conformational changes in the capsid and portal proteins, triggering the dissociation of the terminase complex and the attachment of the tail machinery to retain the packaged DNA [10,11]. The scaffold plays important roles in the formation of the portal and the assembly of the procapsid of phages, herpesviruses [12,13], and adenoviruses [14]. Absence of the scaffold prevents proper assembly of the coat protein and incorporation of the portal and other minor capsid proteins into the procapsid [8,13,15–19].

The structures of some procapsids or assembly intermediates of tailed phages and herpesviruses have been resolved. Cryo-electron microscopy (cryo-EM) structural analyses [20,21] revealed structure of helix-turn-helix motif on the inner surface of the P22 procapsid, which is consistent with the solution structure of the coat protein-binding domain of the P22 SP by nuclear magnetic resonance analysis [22]. Cryo-EM analyses of the procapsids of phages T7, SPP1, and 80α unveiled structures on the inner surface of the coat that were consistent with the coat protein-binding domain of the SP [23–25]. A recent study establishing the cryo-EM structure of the phage HK97 procapsid indicated that the portal was surrounded by scaffolding domains, which formed 10 trimeric coiled coils; however, the rest of the scaffold structure remains unresolved [26]. A study investigating the structure of the B-capsid of human cytomegalovirus (HCMV; family: Herpesviridae) revealed a short fragment of the scaffold in the portal's hydrophobic pocket, with the remaining scaffold appearing as featureless layers [27]. The unresolved scaffold structures in procapsids may be attributed to the scaffold's flexibility and asymmetrical distribution. Recent structural analysis of the phage phiBB1 procapsid indicated that 10 SP subunits bound to the portal, forming an inner shell within the tubular section of the procapsid coat [28]. The aforementioned structures provide valuable insights into scaffold architecture and its interactions with the coat and portal proteins. However, because scaffold structures remain incomplete and biochemical data remain limited, the mechanisms underlying SP-assisted procapsid assembly and SP release during phage maturation remain unclear.

Phage P22 infects *Salmonella enterica* serovar Typhimurium, a primary enteric pathogen of humans and animals. The P22 SP was the first SP ever discovered, and the assembly of P22 has been extensively characterized biochemically [6,12,15,16]. Thus, P22 is an ideal model for studying the structural mechanisms underlying SP-assisted procapsid assembly. The P22 procapsid consists of an icosahedral coat comprising 415 coat protein (gene product [gp] 5) subunits, a dodecameric ring comprising portal protein (gp1) subunits located at a unique capsid vertex [20], and an internal core comprising approximately 250–300 SP (gp8) subunits and 3 ejection proteins [15,29,30]. The P22 SP, which is predominantly α-helical, has a molecular weight of 33.6 kDa and is composed of 303 amino acid residues [29,31]. During DNA packaging, the P22 procapsid undergoes structural changes to form the mature virion [20]. The SP is expelled from the procapsid, possibly as DNA enters [20]. Concurrently, the tail machinery attaches to the portal [32]. Studies have resolved the structure of the P22 procapsid and revealed the U-shaped coat-binding domain of the SP within the inner procapsid [20,21]. However, the complete structure and organization of the SP remain unknown, likely because of its asymmetrical distribution and flexible arrangement.

In this study, we present the structure of the SP gp8 in complex with the dodecameric portal in the P22 procapsid at subnanometer resolution. The AlphaFold2-predicted gp8 structure with minor model refinement has good fit with our structure; thus, we could build a pseudo-atomic model of the SP–portal complex. Our structure suggests that gp8 presents as trimers and tetramers, forming a dome-like structure on the portal. The trimers interconnected with two neighboring portal subunits. Our findings clarify how the SP–portal complex orchestrates procapsid assembly and highlight the mechanisms underlying SP assembly and release during procapsid assembly and virion maturation.

## Results

### Structural determination of P22 procapsid and mature virion

Bacteriophage P22 particles were propagated in *Salmonella typhimurium* cells. Phage lysate from 2 L of bacterial culture was purified through density gradient centrifugation, which yielded two visible bands containing P22 particles. Cryo-EM analysis revealed the presence of procapsids in the upper band and that of mature virions in the lower band (S1A and S1B Fig). Symmetry-mismatch reconstruction of the procapsids and mature virions at resolutions of 9.2 and 8.3 Å, respectively, revealed a unique portal or portal–tail complex in each capsid (Figs 1A–1D and S1C), identical to previously reported structures [20,33,34].

### Structures of the portal in the procapsid and the portal–tail complex in the mature virion

In the mature virion, the portal–tail complex is present at a unique vertex inside the DNA-filled capsid. The portal has a canonical 12-fold structure that is formed by 12 copies of the 725-residue portal protein gp1 arranged around a central tunnel (Fig 1A, 1B, and 1E). Further local reconstruction of the portal by applying 12-fold symmetry yielded the structure of the portal at a resolution of 3.2 Å (Figs 1A, 1B, S1D, and S2), which was identical to those previously reported structures of the portal in the P22 mature virion [33,34] including a lower resolution helical barrel structure, flexible tunnel loop, N-terminal, and top regions in the wing (residues 1–5, 422–442, and 480–487, respectively). The tail structure (Fig 1A and 1B), attached to the portal through interactions with 12 copies of the adaptor protein gp4, was also identical to previously reported P22 tail structures [33,34].

In the procapsid, the portal is located at a unique vertex inside the DNA-free shell (Fig 1C and 1D). Asymmetric reconstruction of the portal at a resolution of 8 Å revealed its overall 12-fold symmetric structure, which was similar to a previously reported lower-resolution structure of the procapsid portal (EMD-1828) [20]. Local reconstruction with C12 symmetry improved the resolution to 3.0 Å (Figs 1D, S1D, and S3). The portal protein gp1 comprises five domains (Fig 1F): the clip (residues 344–398), stem (residues 320–343 and 399–416), wing (residues 6–319 and 417–524), and crown (residues 525–600). The C-terminal helical barrel domain (residues 601–725) appeared unstructured (Fig 1D and 1F). Structural comparison of the procapsid and mature virion portals revealed major differences in the following domains (Fig 1G and S1 Movie): (1) the top region of the wing domain, which interacted with the SP complex (described below); (2) the stem and clip domains, which likely interacted with the terminase complex motor during DNA packaging; and (3) the helical barrel domain, which was absent in the procapsid.

### Overall structure within the procapsid

The density map of the procapsid at a low resolution of 15 Å (displayed at 1.8 times the standard deviation of the density values) revealed two dome-like structures on the top region of the portal (Figs 2A, 2B, and S4A). These dome-like structures, which were sparsely connected

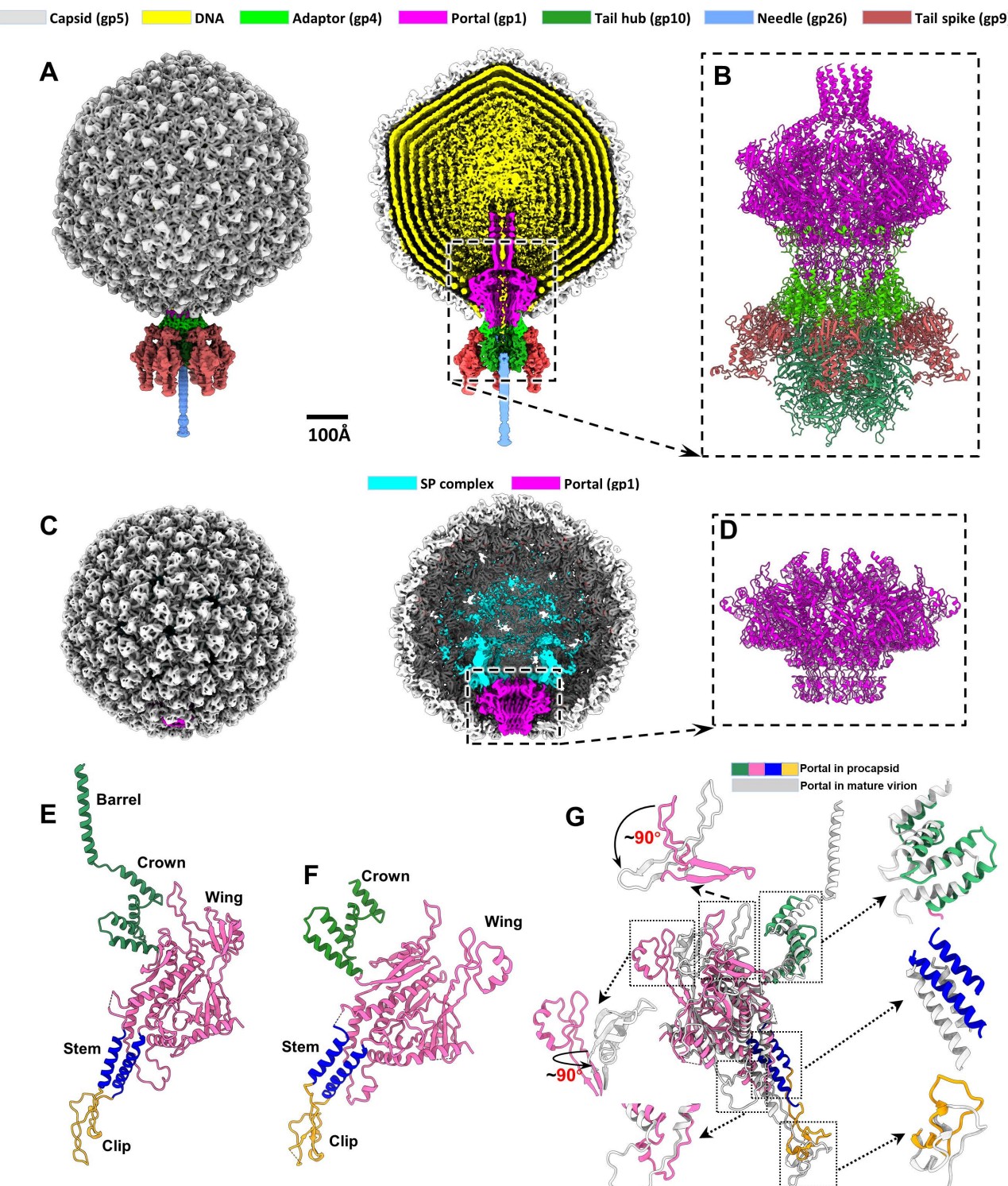

**Fig 1. Structural comparison of the portal–tail complex in the P22 mature virion and the procapsid.** (A) Overall and cut-open views of the P22 mature virion (EMD-61453). The 100-Å scale bar applies to panels A and C. (B) Zoomed-in view of an atomic model of the portal–tail complex in the mature virion (PDB ID: 9JG6). (C) Overall and cut-open views of the P22 procapsid (EMD-61454). (D) Atomic model of the portal in the procapsid (PDB ID: 9JGA). (E) Ribbon model of the portal protein gp1 in the mature virion. (F) Ribbon model of the portal protein gp1 in the pro-capsid. (G) Structural comparison between the portal proteins in the procapsid and mature virion.

by column-like elements (Figs 2B and S4A), were mostly unresolved, whereas the region surrounding the portal was better resolved (Figs 2A, 2B, and S4A). We speculated that these domes were formed by multiple subunits of the SP and ejection proteins. The lower resolution may be attributable to structural flexibility within the procapsid. The structures of the domes and portal were analyzed using the structure-analysis software program ChimeraX [35] at different display levels. If a higher display level is set, the regions with more flexible organization in the phage structure would disappear in the density map; otherwise, these regions would be present. To improve the resolution of the regions of interest, we performed local reconstruction, analyzing the flexible structure and organization of the SP gp8 subunits on the portal.

## Structure of the SP complex around the portal crown domain

When the display level was increased beyond 2.4 times the standard deviation, the two domes appeared unstructured, likely because of their flexibility, but the structures surrounding the portal were visible (Figs 2C, S4B, and S4C). The density map revealed nine nodule-like structures located on nine of the 12 symmetrical wing domains (Fig 2C–2E). These nodules were organized into two groups of four (GOF) nodules and a single nodule (Fig 2C–2E). Three column-like structures, each interacting with two neighboring nodules, were observed on the GOF nodules (Fig 2E). From each column, another nodule extends outward (referred to as a side nodule) through a stalk-like structure (Fig 2C, 2E, 2F, and 2K).

Local reconstruction of the nodules and columns improved the resolution to approximately 8 Å. The nine nodules, which were essentially identical, were formed by α-helices (Fig 2H and 2J), and each column was formed by a three-helix bundle (Fig 2F). We speculated that the nodules and columns were formed by the SP gp8. Therefore, the structure of gp8 was predicted using AlphaFold2 [36]. The predicted structure comprised five domains (Fig 2G): N-loop (residues 1–57), base (residues 58–84 and 163–245), column (residues 85–162), C-loop (residues 246–267), and hook (residues 268–303). The main body (base and column domains) of the predicted gp8 model with minor refinement had good fit with the density maps of the GOF nodules and columns (Figs 2F and S5). The gp8 subunits formed the GOF nodules and columns through complex interactions. Each nodule comprised two copies of the base domain (base dimer) interacting with each other (Fig 2H–2J). We referred to the bases near the portal axis as inner base and those far from the axis as outer base (Fig 2H–2J). The two base domains likely interacted through electrostatic interactions. Residues Arg240, Glu243, and Arg244 of the inner base domain were located close to residues Arg240 and Glu243 of the outer base domain (Fig 2J). Furthermore, each nodule interacted with the portal through the stacking of the inner base on a loop (residues 495–504) of the portal wing (S6 Fig).

Local reconstruction of the side nodule and stalk-like structure revealed that the side nodule comprised two copies of the base domain, whereas the stalk comprised the column domain, which connected one of the two base domains in the side nodule (Figs 2F and S5). On each GOF, each column connecting two neighboring nodules consisted of a three-helix bundle (Fig 2F and S2 Movie). The three α-helices were contributed by the column domains of the inner base-associated gp8, the outer base-associated gp8 from the neighboring nodule, and the side nodule-associated gp8 (Fig 2F and 2K). These three SP subunits interacted through the three-helix bundle, forming an SP trimer (Figs 2F, 2K, and S5). The gp8 subunits interacted with the neighboring portal subunits (Fig 2K).

## Structure of the SP complex above the portal crown domain

When the density map of the portal and domes at a resolution of 15 Å was displayed at 1.9 times the standard deviation, the side view of the GOF nodules revealed that the outer dome

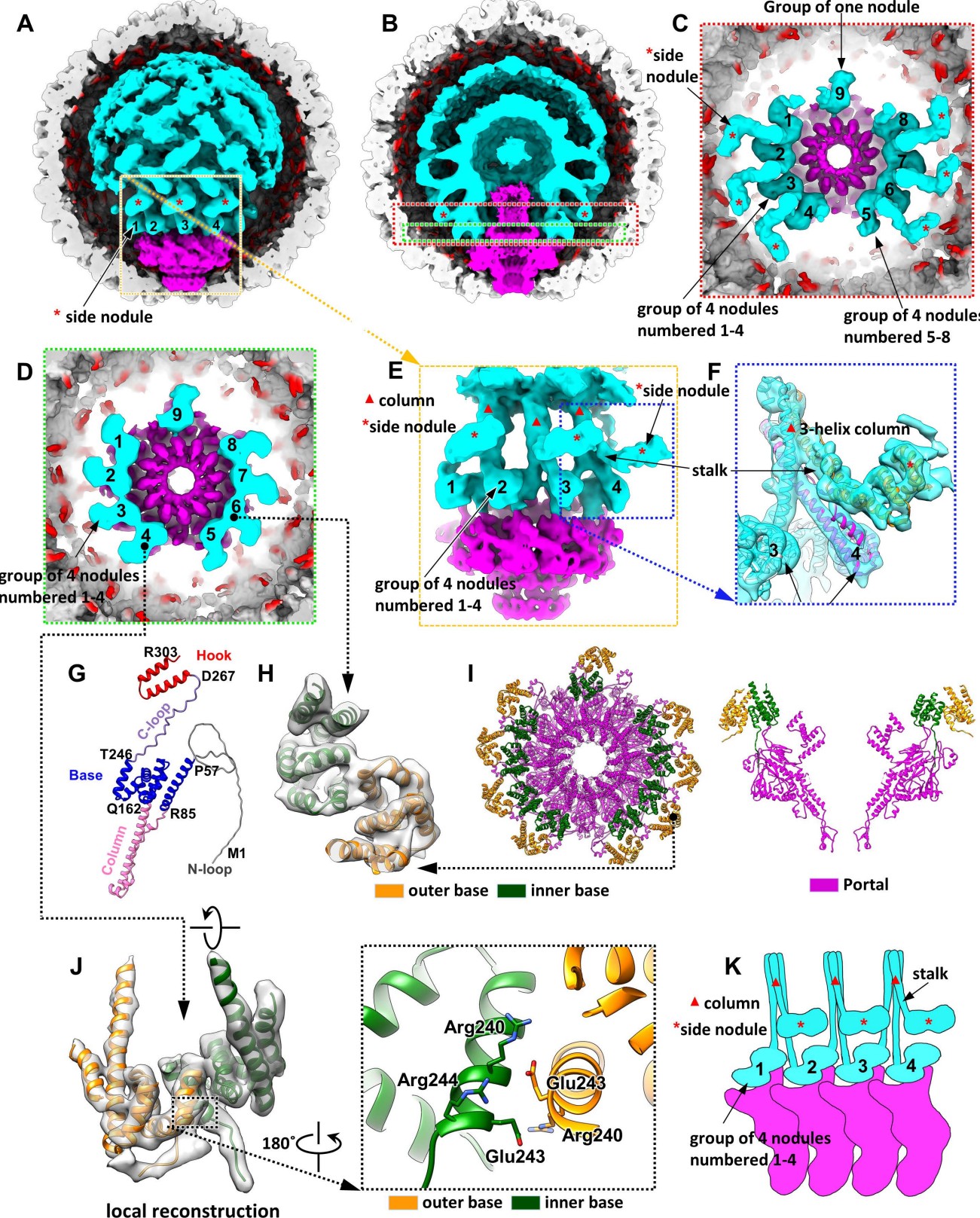

**Fig 2. Structure of the P22 procapsid. (A)** Structures of the SP complex (cyan) and portal (magenta) within the procapsid (EMD-61454). The front half of the capsid is not shown. The SP hook domains anchored to the inner capsid surface are indicated in red. **(B)** Cut-open view of the SP complex.

**(C)** Slab-view of the red box from the top in panel (B). Each side nodule is marked with an asterisk. The nine nodules on the portal are numbered. **(D)** Slab-view of the green box from the top in panel (B). The nodules on the portal are numbered. **(E)** SP complex and portal viewed from the left side of panel (C). Each column is marked with a triangle. Each side nodule is marked with an asterisk. The nodules on the portal are numbered. **(F)** Local reconstruction of the density map in the blue box in panel €. The side nodule in the middle is removed for clarity. The density map is superimposed on three SP models (EMD-61455; PDB ID: 9KYY). **(G)** Ribbon view of the AlphaFold2-predicted SP model, with SP domains in different colors. **(H)** Zoomed-in view of a nodule density map superimposed on two copies of the base domains of the predicted SP model (EMD-61452; PDB ID: 9KYV). **(I)** Left: atomic model of the portal (magenta) and the nodules (PDB ID: 9KYV, 9JGA). The inner SP base domains are in cyan, and the outer base domains are in orange. Right: a slab-view of the left view. **(J)** Zoomed-in view of a nodule density map superimposed on two copies of the base domains of the predicted SP model (PDB ID: 9KYV). The color scheme of the models is identical to that in panel (I). **(K)** Schematic of panel (E). The four nodules (group of 4) are numbered 1–4. Each nodule is formed by a dimer of the SP base domain. Each column (marked with a red triangle) is a three-helix bundle (three SP column domains). The side nodules (base dimer) are marked with asterisks.

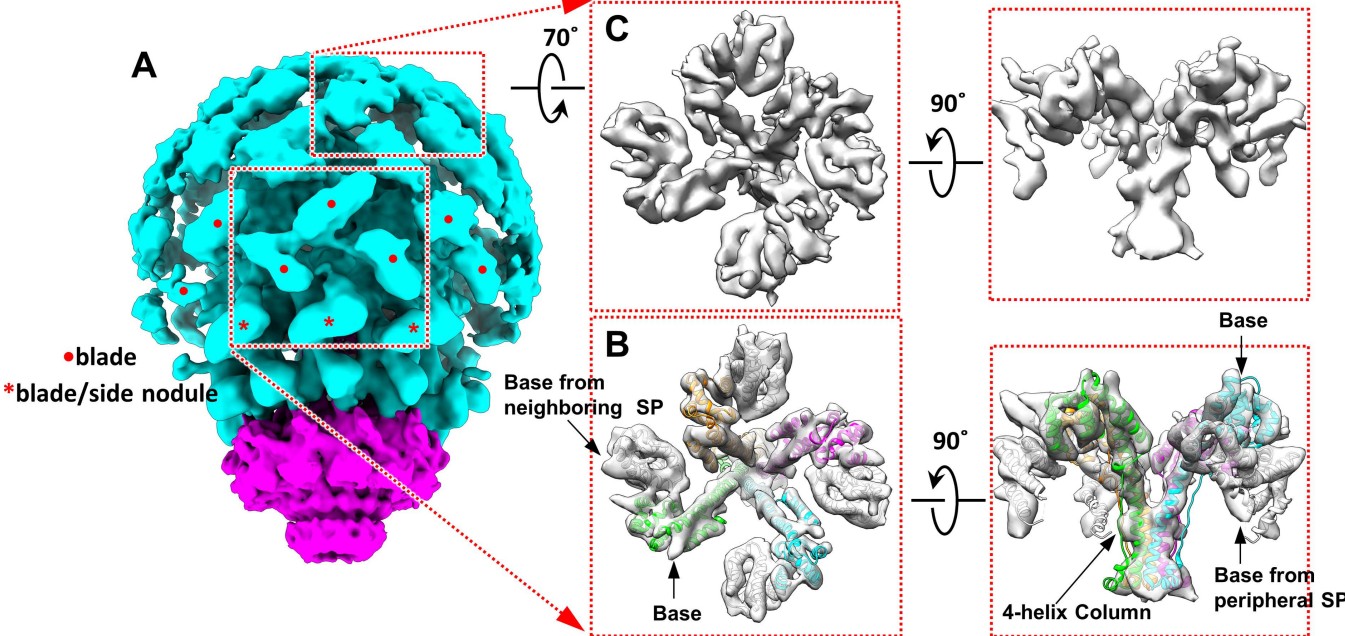

**Fig 3. Structure of the SP complex and portal. (A)** Structure of the SP complex and portal within the capsid (EMD-61454). The capsid is not shown. A central windmill-like structure is highlighted in a red dashed box. Each side nodule, which contributes to a windmill blade, is marked with an asterisk. The remaining blades are marked with red dots. The structure of the SP-portal complex is tilted approximately 20°along the horizontal axis relative to that presented in Fig 2A. **(B)** Local reconstruction of the density map (transparent gray) of the SP tetramer at a resolution of 6.9 Å, superimposed on four copies of the gp8 models (magenta, cyan, green, and orange). The outermost part of each blade is contributed by the base domain (gray model superimposed) of a neighboring gp8 subunit (EMD-61456; PDB ID: 9KYX). **(C)** Local reconstruction of the top region of the dome, indicating that this region is also formed by the SP tetramers.

contained three neighboring four-blade windmill-like structures (Fig 3A and 3B). The four blades of each windmill-like structure in the outer dome were connected to the inner dome through a column, with one blade contributed by a side nodule. Given the structural similarity between the blades and the nodules, we speculated that each blade in the windmill-like structure comprised two base domains. To confirm this notion, we performed local reconstruction of the windmill-like structure. The results indicated that each windmill-like structure comprised four gp8 subunits, constituting a gp8 tetramer. Each blade comprised a base dimer. The outermost part of each blade was not part of the gp8 tetramer but was instead contributed by the base domain of a neighboring gp8 subunit (Fig 3B and S2 Movie). The gp8 tetramer was stabilized through a four-helix bundle formed by the four-column domains (Fig 3B). The

tetramers interacted with other gp8 subunits in neighboring tetramers through electrostatic interactions between the base domains.

Local reconstruction of the top region of the outer dome revealed that this region comprised gp8 tetramers (Fig 3C). The nodules, side nodules, and windmill blades were all dimers of the SP base domain. The column-like structures connecting the outer and inner domes consisted of helix bundles formed by three or four α-helices of the SP column domain (Figs 2F, 3B, and S7). In total, 96 base dimer models could be fitted into the overall density map of the outer dome (S8 Fig). This finding aligns with those of biochemical studies reporting 250–300 copies of the SP within the P22 procapsid [15,29,37]. Only 96 base dimers (192 base domains) could be fitted into the outer dome, likely because of the overall flexibility of the dome's top region.

In summary, the SP complex is composed of SP trimers and tetramers, which are formed by SP subunits assembling into three-helix and four-helix bundles, respectively (S2 Movie and S7 Fig). These trimers and tetramers interacted through the SP base domains (base dimers), forming a lattice of base dimers within the outer dome (S8 Fig). The interactions among the three or four SP column domains and those between the base domains mediated the multimerization of the SP subunits.

### Interactions among the SP complex, procapsid shell, and portal

The SP interacts with the coat protein through a U-shaped hook domain [20–22]. In our asymmetrical reconstruction of P22, the hooks were distributed across the inner surface of the procapsid shell, which is consistent with previously reported icosahedral reconstructions [20,21]. In our reconstruction, the distribution of the hooks interacting with the five hexamers surrounding the portal was asymmetric (Fig 4A). The weak density of these hooks suggests partial occupancy on each coat protein subunit of the procapsid shell, consistent with biochemical studies revealing approximately 250–300 copies of the SP and 415 copies of the coat protein within the P22 procapsid [15,29,37]. The hooks between the portal and the coat protein mediate the interactions between them (Fig 4B and 4C). In our previous icosahedral analysis of the P22 procapsid [21], each hook interacted with the inner capsid primarily through electrostatic interactions mediated by Arg293 and Lys296 in the hook and by Asp14, Glu15, and Glu18 in the N-arm domain of the coat protein gp5. Superimposition of the portal and hook models onto the asymmetric reconstruction of the procapsid indicated that the asymmetrical interactions between the hooks and portal are also electrostatic.

To explore the interactions between the C-loop of the SP and the portal, we performed local reconstruction of the C-loop associated with the inner base. The C-loop, which extends from the inner base domain and interacts with a hook on the portal surface, is primarily located in a cleft between two adjacent wing domains of the portal subunits (Fig 4C–4E). The C-loop interacts with Val474, Leu292, Tyr191, Phe187, and Met179 in the wing domain of a portal subunit and with Ile266 in the wing domain of the adjacent portal subunit (Fig 4C–4E). During DNA packaging, a conformational change in the portal may disrupt these interactions, resulting in the dissociation of the C-loop.

### Discussion

A biochemical study demonstrated that SP facilitates portal ring formation by interacting with portal monomers [12]. Although portal monomers can polymerize into dodecameric rings, they do so less efficiently [38]. Our findings suggest that the portal and SP complex form the SP–portal complex within the P22 procapsid. Nine nodules are formed by the base dimers interacting directly with the portal's wing domain. The reason why only nine nodules occupy the 12 symmetrical positions on the portal remains unclear. A previously reported structure of the

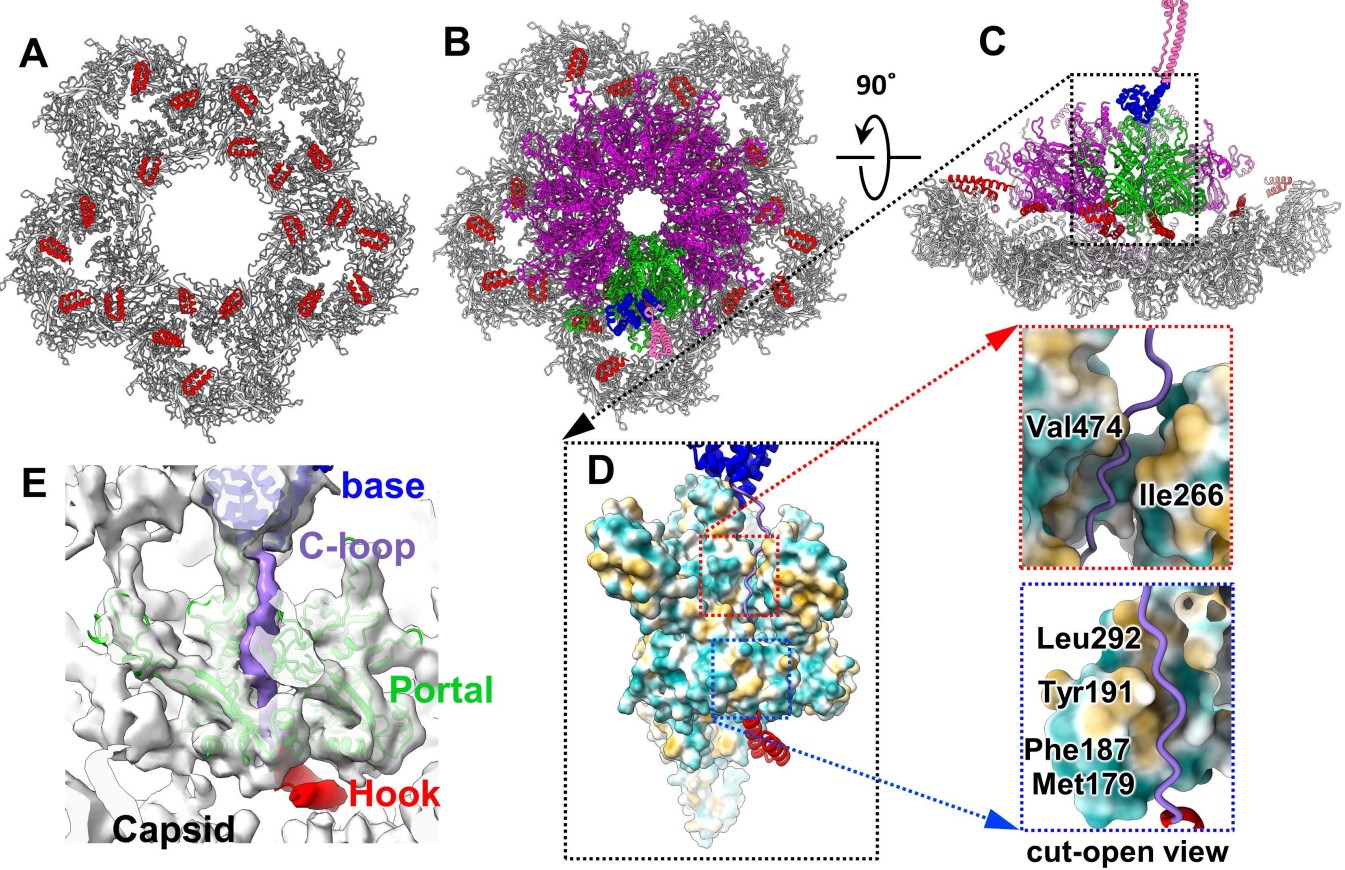

**Fig 4. Interactions among the SP, portal proteins, and major capsid proteins. (A)** Distribution of the hooks (red) on the inner surface of the capsid surrounding the portal (the portal is not shown) (PDB ID: 8I1V). **(B)** The portal and a copy of the inner base-associated SP are presented (PDB ID: 8I1V, 9KYW). Two of the 12 portal subunits interacting with the SP are indicated in green, whereas the remaining subunits are indicated in magenta. The color scheme for the SP domains matches that in Fig 2G. **(C)** Side view of panel (B). **(D)** Hydrophobic interactions between the portal protein (surface view) and the C-loop of the SP (ribbon view). **(E)** Models of the SP and portal protein superimposed onto the density map (EMD-61461; PDB ID: 9KYW).

P22 procapsid [20] exhibited a similar organization of the base and column domains. However, compared with nine nodules in our structure, 10 nodules occupied the portal's wing domain in the aforementioned structure [20]. If a lower display level is set, 10 nodules can be observed in our structure (S9 Fig). Therefore, the associations of the nodules with the portal are flexible.

Each SP trimer interacts with two neighboring portal subunits (Fig 2K), and the SP's C-loops are embedded in the cleft between these subunits (Fig 4D and 4E). This finding is consistent with that of a study indicating that SP interacts with portal monomers to catalyze portal ring formation [12]. A U-shaped hook domain is present at the C-terminal end of the SP; several copies of hook domain anchored on the portal surface mediate the interaction between the portal and coat (Fig 4B and 4C). We propose that the SP and portal protein form a nucleation complex, which is essential for the assembly of the icosahedral procapsid. First, the hooks on the portal recruit and anchor coat protein subunits, forming five hexamers around the portal (Fig 4B and 4C). Then, assembly proceeds through the addition of SP subunits to the SP-portal complex, forming the portal's peripheral region of the outer dome.

The inner dome likely comprises three ejection proteins gp7, gp16, and gp20 because (1) the SP may be required for the assembly of gp16 and gp20 into the procapsid [19,39], and the

addition of physiological quantities of gp16 increases the rate of procapsid assembly [40]; (2) the three ejection proteins may interact with each other [41]; and (3) the inner dome is located around the portal helical barrel domain (although mostly disordered) (Fig 2B), where the three ejection proteins are present in the P22 mature virion [41]. The portal's helical barrel is fully established after the maturation of the phage P22 [20,33,42]. Our structure indicates that the inner end of the helix bundle (SP column) serves as the docking site for the ejection proteins. Moreover, the inner dome, in turn, enhances the stability of the SP layer. The organization of the ejection proteins and SP indicates that the ejection proteins coassemble with the SP during procapsid assembly. The inner dome does not directly interact with the portal in the procapsid. Indeed, the ejection proteins can assemble into the procapsid even in the absence of the portal protein [40].

During assembly, each SP subunit likely binds to a coat protein subunit through the hook domain. Therefore, the two domes and the coat assemble simultaneously; otherwise, the long C-loops projecting from the outer dome would become entangled. Through this simultaneous assembly, adjacent SP subunits bring coat protein subunits into proximity, and the flexible C-loops provide sufficient freedom for the coat protein subunits to interact through their intrinsic contacts. No evidence suggests that coat protein subunits form hexamers or pentamers before their incorporation into the procapsid [6,12,43,44]. Instead, the SP complex likely induces coat protein oligomerization, facilitating the formation of hexameric or pentameric coat protein oligomers. Some coat protein subunits may be secured in place by preassembled subunits without assistance from the SP. Given the size of the outer dome (diameter: approximately 380 Å), coat protein subunits cannot polymerize into an aberrant $T = 4$ procapsid shell (diameter: approximately 390 Å). $T = 4$ shells assemble in the absence of the SP [45]. The domes and procapsid coat continue to assemble at the edge of the growing outer dome and coat until a complete procapsid is formed. The building blocks of the assembly could be the SP monomers, dimers, or tetramers, given that the P22 SP exists in solution as a mixture of these forms [46]. The SP dimer primarily forms through interactions between two copies of the same α-helix (residues 231–245) within the base domains; this explains why a recombinant SP fragment (residues 238–303) weakly self-associates into dimers [22]. The SP tetramer primarily forms through interactions of the C-terminal region (residues 122–136) within the column domain; this explains why a recombinant SP fragment (residues 141–303) does not form tetramers [47].

The portal undergoes conformational changes during DNA packaging and phage maturation (Fig 1G), triggering the dissociation of the C-loops of SP subunits from the portal. These processes likely lead to the disassembly of the dome architecture and the expulsion of the SP. The SP is expelled through central openings in the coat hexamers, creating space for viral DNA. The three ejection proteins (gp7, gp16, and gp20), which likely form the inner dome surrounding the portal's helical barrel in the P22 procapsid, may be loosely clustered around the helical barrel in the mature virion [41].

The 57-residue N-loop of the P22 SP was not resolved in our procapsid structure. A study demonstrated the formation of virion-like, DNA-containing particles even in the absence of the N-loop. However, in such cases, the SP remained inside the capsid after DNA encapsidation, resulting in a shorter encapsidated DNA molecule than the wild-type genome [39]. These findings implicate the negatively charged N-loop in SP release during DNA packaging. In addition, the mutant virion particles contained a substantially reduced amount of gp16 [39], suggesting that the N-loop facilitates the assembly of gp16.

In addition to phage P22 [6,12], phages such as T4, phi29, and SPP1 use the portal as a nucleator for procapsid assembly; interaction between the portal and the SP is necessary for the incorporation of the portal into the coat [4,48–52]. Building the capsid around the

SP–portal complex ensures the incorporation of a single portal at a unique capsid vertex [53]. The overall structure of the SP in most phages remains unresolved, likely because of its flexible organization, asymmetrical distribution, and transient presence in the procapsid. A recent cryo-EM structure of the phiBB1 procapsid, a prolate icosahedron, provided the SP structure within the procapsid [28]. Ten SP subunits interacted with 10 of the 12 portal subunits (S10A–S10C Fig). In addition, an inner shell comprising SP subunits was observed beneath the tubular section of the coat. The central region of the predicted phiBB1 SP model aligned well with the density maps of the SP subunits [28]. The well-resolved SP structure in the phiBB1 procapsid may be attributable to the 1:1 stoichiometric ratio [28] of the SP and coat protein. The overall structure of the phiBB1 SP resembles that of the P22 SP (S10A and S10B Fig). Each SP subunit is associated with a portal subunit through interactions between the base domain and the top of the wing domain (S10C Fig). Interactions among the SP subunits beneath the tubular section are mediated by the column domain (two-helix bundle) and base domain (dimer), similar to those observed in the P22 SP (S10D and S10E Fig). The cryo-EM structure of the phage HK97 procapsid, where an N-terminal scaffolding domain is covalently linked to the coat protein, revealed that the portal is surrounded by scaffolding domains forming of 10 trimeric coiled coils; the rest of the scaffolding domain was unresolved [26]. The trimeric coiled coils comprise three long α-helices, with each corresponding to the P22 SP column domain. Notably, all coat protein-binding regions of these SPs (or domains) are also located at their C-termini [22,26,28].

Herpesviruses, the eukaryotic counterparts of tailed dsDNA phages, have a portal structure similar to that of tailed bacteriophages [10]. In herpes simplex virus 1, the SP–portal complex is thought to initiate procapsid formation or be incorporated early during assembly [5,13]. In this virus, major capsid proteins bind to 25 C-terminal residues of SPs encoded by UL26 and UL26.5 [54,55]. A recent study investigating the cryo-EM structure of the HCMV B-capsid indicated the layered structure of the scaffold [27]. The B-capsid represents the pre–DNA-packaging form of herpesviruses [56]. Twelve short loop-helix-loop fragments of the scaffold surround the HCMV portal crown region, with each fragment accommodated within a hydrophobic cavity formed by two neighboring portal subunits [27].

On the basis of the structures of the portal and SP complex within the P22 procapsid and structural comparisons between P22 and other dsDNA viruses, we drew the following conclusions. First, the main body of the SP comprises a nodule-like base domain and a long-helix column domain. Second, within the procapsid shell, SP subunits interact by forming two-, three-, and four-helix bundles with the long-helix column domains and by dimerizing through the base domains, forming a spherical disordered layer (or layers), which serves as a scaffold for the assembly of major capsid proteins. Finally, the SP complex interacts with the portal through the base domains, and the structures adjacent to the portal are more ordered than those distal from the portal. The structural similarity among procapsids of dsDNA viruses suggests that tailed phages, herpesviruses, and adenoviruses employ similar mechanisms for procapsid assembly.

## Materials and methods

### Purification of bacteriophage P22 particles

The P22 procapsid and mature virion particles were purified as described previously [21]. In brief, the *S. typhimurium* strain was grown in Luria-Bertani broth (tryptone, 10 g; yeast extract, 5 g; and NaCl, 10 g/L) for 5 h at 37 °C. P22 phages (ATCC-97,541-B1) were propagated using *S. typhimurium* cells for 4 h at 37 °C. The bacteria were separated and the supernatant was collected through centrifugation at 6,000*g* for 15 min at 8 °C. The bacterial cells were lysed

with 50% chloroform, and the cell debris was removed through low-speed centrifugation. The supernatant was concentrated through overnight precipitation using polyethylene glycol 8,000 (10% [w/v] polyethylene glycol in 1 M NaCl) in an ice-water bath. The precipitated phage particles were resuspended in phage buffer (10 mM Tris-HCl and 1 mM $MgCl_2$, pH 7.4) and were then purified by two rounds of cesium chloride (CsCl) density gradient centrifugation. After the first round of centrifugation using 1.4 and 1.6 g/mL CsCl cushions at 100,000$g$ for 2 h at 8 °C, two distinct bands for phage particles were separated. The upper and lower bands were further purified through a second round of centrifugation with 1.4 and 1.6 g/mL, respectively. Subsequently, the purified P22 particles from the two bands were dialyzed overnight against phage buffer. Finally, the purified P22 particles were stored in an ice-water bath for subsequent processing.

## Cryo-EM and data collection

Three μL of the purified P22 particles, which were obtained from the two bands, were carefully pipetted onto an amorphous nickel-titanium alloy grid with carbon film previously glow-discharged for 30 s. The grid was blotted with filter paper for 3.5 s at 100% relative humidity and 8 °C, and then rapidly frozen by plunging into liquid ethane using a Thermo Fisher Scientific (TFS) Vitrobot. Subsequently, the grids were observed under a TFS Krios G3i transmission electron microscope operated at a voltage of 300 kV, which was equipped with a Gatan imaging filter and a K3 direct electron detector for recording cryo-EM movies. The Gatan imaging filter was utilized with a slit width of 20 eV.

Image collections for the purified P22 particles from the lower and upper bands were performed automatically using the TFS EPU software at magnifications of 53,000× and 64,000×, corresponding to pixel sizes of 1.36 and 1.06 Å, respectively. Finally, a total of 4,668 movies (fractioned into 32 frames) were collected for P22 particles from the upper band, and 3,733 movies were obtained for those from the lower band. The total dose for each movie was 35 e⁻/Å², and the defocus values ranged from 1.8 to 2.2 μm.

## Image processing and icosahedral reconstruction

The defocus and astigmatism values of each micrograph were calculated using our software *dfsearch*. The P22 particles were automatically boxed using ETHAN software [57], Moreover, the procapsid and mature virion particles were manually selected for image reconstruction. The orientations and centers of these two types of particle images were determined by our software *icosprocess* [58], which is based on the common-line algorithm [59,60]. Icosahedral symmetry was imposed for the reconstruction of the head structures of the two types of particles by using our reconstruction software ISAF [61].

## Symmetry-mismatch and local reconstruction

On the basis of the orientation and center parameters of each particle image obtained from the icosahedral reconstruction, we reconstructed the intact asymmetric structure of phage P22 by using our symmetry-mismatch reconstruction software package *icosprocess_mismatch* [11,62]. Briefly, for each image of the mature particle, we initially localized the unique vertex with the portal–tail by searching the 12 icosahedral vertices that were localized during the icosahedral reconstruction step. Subsequently, a low-resolution structure of the phage incorporating the portal–tail complex was obtained without imposing symmetry. Using this structure as the initial model, the resolution was then enhanced through an iterative process involving projection–refinement–reconstruction steps. In this step, we calculated the correlation coefficients between 60 equivalent icosahedral projection images and the particle image to determine the

best matching asymmetric orientation. The iteration continued until the orientations of all particle images were stabilized and the phage structure (at a resolution of approximately 8 Å) could no longer be enhanced. Next, we proceeded with the segmentation of the portal–tail complex from the phage structure as an initial model and extracted the two-dimensional portal-tail images from the original particle images for local refinement and reconstruction. By imposing 6-fold symmetry, we determined the structure of the portal-tail complex (gp1, gp4, gp9, gp10, and gp26) at a resolution of 3.2 Å.

The P22 procapsid particles in the upper band were reconstructed using the same approach. By using symmetry-mismatch reconstruction, we obtained the procapsid structure containing the portal and SP at a resolution of approximately 9 Å, and the portal of the procapsid at a resolution of 3.0 Å was determined using local reconstruction, for which 12-fold symmetry was imposed. On the basis of the orientation and center parameters of the procapsid that were obtained through symmetry-mismatch reconstruction, the structures of the portal and SP complexes were locally reconstructed, including the nine nodules (dimers of the SP base) on the portal wing. Reconstruction was conducted at a resolution of 20-Å without imposing symmetry using 22,169 images of procapsid particles. The nine nodule structures located on the 12-fold symmetrical wing domains were then averaged. To identify the 12 potential locations of the nine nodules in each procapsid image, we imposed 12-fold symmetry and considered the orientation and center parameters of the procapsid. We boxed out 12 possible nodule images based on the 12 potential locations of the nodules. To exclude the three incorrect locations in each procapsid image, the 12 potential nodule images were compared with a nodule template by using a phase residual (PR) distribution-based selection method, which uses the average amplitude-weighted PR of the common lines between each particle image and the template images as a measure for particle selection. This method identifies particle images for reconstruction by setting a threshold at the valley of the PR distribution curve and selecting those within the Gaussian region at the lower end. A total of 199,080 nodule images were derived from the initial dataset of 22,169 procapsid particle images. A 3D mask model was generated to iteratively refine the orientation and center of the nodule structures in the procapsid particle images using our local_refine and local_reconstruct software, which were based on a model-based algorithm [63]. Finally, the nodule structures were reconstructed at a resolution of 4.9 Å.

On the basis of the orientation and center of the nodules, we expanded the reconstruction region to include the SP trimer because one base in the nodule belongs to the SP trimer (Fig 2E, 2F, and 2K). Through iterative local refinement and reconstruction, we obtained the structure of the SP trimer at a resolution of 6.9 Å. The SP tetramer was reconstructed using the same approach as that used for nodule reconstruction. A total of 33,180 SP tetramer images were selected from the 22,169 procapsid images by using the PR distribution-based selection method. Then, the SP tetramer was reconstructed at a resolution of 6.9 Å without imposing symmetry. Additionally, 21,884 particles were selected for the reconstruction of the C-loop structure at a resolution of 6.7 Å. The resolutions of all reconstructions were estimated using Fourier shell correlation criteria with a cut-off of 0.143. Our software package can be downloaded from the following website: https://doi.org/10.5281/zenodo.8378566.

## Model building

Using the software *COOT* [64], we manually built the atomic models of the portal–tail complex (the proteins gp1, gp4, gp9, gp10, and gp26) on the basis of our cryo-EM density map of the P22 mature virion, and the protein gp1 on the basis of our density map of the P22 procapsid. The gp8 structure predicted by AlphaFold2 [36] was refined on the basis of our gp8

complex density map using software *COOT* [64]. The model with minor refinement could be accurately fitted into the SP density map using *UCSF Chimera* [65]. Furthermore, we refined the models (gp1, gp4, gp9, gp10, and gp26 in the mature virion and gp1 in the procapsid) through real-space refinement, which was implemented in *Phenix* [66]. The refinement and validation statistics are presented in S2 Table. All figures were prepared using *UCSF Chimera* [65] and *ChimeraX* [35].

## Supporting information

**S1 Fig. Cryo-EM images and Fourier shell correlation curves. (A)** Two bands for phage particles separated through CsCl density gradient centrifugation. **(B)** Cryo-EM images of P22 particles from the upper (left) and lower (right) bands. **(C)** Structural resolutions of the overall symmetry-mismatch of the mature virion and procapsid as well as the dimer, trimer, tetramer, and C-loop of the SP in the procapsid were estimated based on Fourier shell correlation (FSC) criteria by using the gold-standard procedure. **(D)** Structural resolutions of the portal in the P22 procapsid and portal–tail complex in the P22 mature virion. The cryo-EM density maps (EMD-61452, EMD-61453, EMD-61454, EMD-61455, EMD-61456, EMD-61457, EMD-61460, EMD-61461) have been deposited in the EM Data Bank.
(TIF)

**S2 Fig. Density map of the portal–tail complex of the mature virion.** Zoomed-in views of the density maps of the portal, adaptor, tail hub, tail spike in the mature virion (transparent gray) superimposed on models (EMD-61457; PDB ID: 9JG6).
(TIF)

**S3 Fig. Density map of the procapsid portal.** Zoomed-in views of the density maps of three segments (transparent gray) superimposed on models (EMD-61460; PDB ID: 9JGA).
(TIF)

**S4 Fig. Cut-open views of the procapsid density map (EMD-61454) at a resolution of 15 Å at different display levels in software UCSF ChimeraX.**
(TIF)

**S5 Fig. Fitting the Alphafold2-predicted model of gp8 into the density map of the gp8 trimer. (A)** Fitting, without refinement, of three copies of predicted gp8 models (cyan, magenta, and orange) and three base domains (gray) from neighboring gp8 subunits into the density map of the scaffold trimer. The column domains did not fit well into the density map, but the base domains exhibited a good fit. **(B)** After minor refinement, the predicted gp8 models fit well into the density map of the gp8 trimer (EMD-61455; PDB ID: 9KYY).
(TIF)

**S6 Fig. Interactions between the nodule and portal.** The nodule interacts with the portal through the stacking of the inner base domain on a loop (residues 495–504) of the portal wing (PDB ID: 9KYW).
(TIF)

**S7 Fig. The SP complex is formed by SP trimers and tetramers, which interact with each other through the formation of SP base dimers.** These dimers form the outer dome. **(A)** Structures of the SP complex (cyan) and portal (magenta) within the procapsid (EMD-61454). The front half of the capsid is not shown. **(B)** Zoomed-in view of the SP complex within the yellow box in panel A. **(C)** Zoomed-in view of the SP trimer (EMD-61455; PDB ID: 9KYY) within the blue box in panel B. The middle side nodule that covers the trimer is removed for

clarity. **(D)** Zoomed-in view of the SP tetramer (windmill) (EMD-61456; PDB ID: 9KYX) within the red box in panel A. One of the windmill blades is contributed by a side nodule (asterisked). All blades are formed by dimers of the SP base domain. One of the two bases in each blade is contributed by a neighboring SP subunit. **(E)** Local reconstruction indicates that the top region of the SP complex is also formed by SP tetramers.
(TIF)

**S8 Fig. Fitting of 96 models of the base dimer (cyan) into the density map of the outer dome, which comprises a lattice of base dimers (EMD-61454; PDB ID: 9JGA, 9KYV).**
(TIF)

**S9 Fig. Structural comparison between our structure and a previously reported P22 procapsid structure.** Right: Our structure; the slab-view is identical to that in Fig 2D. Left: EMD-1827, a structure reported by Chen and colleagues in 2011 (PNAS 108:1355–1360). Both structures were filtered to a resolution of 15 Å.
(TIF)

**S10 Fig. Structural comparison between phiBB1 SP and our P22 SP. (A)** Atomic model of the phiBB1 SP (column and base domains are in pink and blue, respectively) superimposed on a corresponding density map (transparent gray) (PDB ID: 8pht). **(B)** Column and base domains of our P22 SP subunit. **(C)** Ten SP subunits (cyan) surround around the portal (magenta) in the phiBB1 procapsid (EMD-17675). **(D)** Presence of SP subunits beneath the tubular section of the phiBB1 procapsid indicates that SP interactions are mediated by the column and base domains (EMD-17674). SP interactions in phiBB1 are similar to those in P22 (E).
(TIF)

**S1 Table. Interactions between the SP hook and portal proteins.**
(PDF)

**S2 Table. Refinement and model statistics.**
(PDF)

**S1 Movie. Conformational changes of portal from procapsid to mature state.**
(MP4)

**S2 Movie. Structures of the SP complex and portal within the procapsid.**
(MP4)

## Acknowledgments

Cryo-EM data collection was carried out at the Kobilka Cryo-EM Center of the Chinese University of Hong Kong (Shenzhen) and the Shuimu BioSciences.

## Author contributions

**Conceptualization:** Lingpeng Cheng, Hongrong Liu.

**Data curation:** Hao Xiao, Jingdong Song.

**Formal analysis:** Wenyuan Chen.

**Funding acquisition:** Hongrong Liu.

**Investigation:** Hao Xiao.

**Methodology:** Jingdong Song, Hongrong Liu.

**Software:** Hao Xiao, Wenyuan Chen, Hao Pang, Jing Zheng, Hao Feng, Lingpeng Cheng, Hongrong Liu.

**Validation:** Wenyuan Chen, Li Wang, Jingdong Song.

**Visualization:** Hao Xiao, Jingdong Song, Lingpeng Cheng, Hongrong Liu.

**Writing – original draft:** Hao Xiao, Lingpeng Cheng, Hongrong Liu.

**Writing – review & editing:** Hao Xiao, Lingpeng Cheng, Hongrong Liu.

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
