## [Editor Report · Decision Letter 0]

17 Sep 2024

Dear Dr Liu, 

Thank you for submitting your manuscript entitled "Structure of the scaffolding protein and portal within the bacteriophage P22 procapsid provides insights into the mechanism of assembly" for consideration as a Research Article by PLOS Biology. Please accept my sincere apologies for the delay in getting back to you with feedback.

Your manuscript has now been evaluated by the PLOS Biology editorial staff, and I am writing to let you know that we would like to send your submission out for external peer review.

Once your full submission is complete, your paper will undergo a series of checks in preparation for peer review. After your manuscript has passed the checks it will be sent out for review. To provide the metadata for your submission, please Login to Editorial Manager (https://www.editorialmanager.com/pbiology) within two working days, i.e. by Sep 19 2024 11:59PM.

Kind regards,

Richard

Richard Hodge, PhD

rhodge@plos.org

PLOS

---

## [Decision Letter · Decision Letter 1]

27 Nov 2024

Dear Hongrong,

Thank you for your continued patience while your manuscript "Structure of the scaffolding protein and portal within the bacteriophage P22 procapsid provides insights into the mechanism of assembly" was peer-reviewed at PLOS Biology as a Research Article. Please accept my sincere apologies for the delays that you have experienced during the peer review process. Your manuscript has now been evaluated by the PLOS Biology editors, an Academic Editor with relevant expertise, and by three independent reviewers. 

In light of the reviews, which you will find at the end of this email, we would like to invite you to revise the work to thoroughly address the reviewers' reports.

As you will see, the reviewers are generally positive about your study but they raise overlapping concerns with the presentation and description of the results and suggest including supplementary movies to make the structures and conformational changes easier to comprehend. In addition, Reviewer #1 raises concerns with the overall level of the reporting in the manuscript and notes that the pseudo-atomic models have not been deposited in a public repository. After discussions with the Academic Editor, we also ask that the software is appropriately documented and made publicly available. Finally, we note that there several errors of fact in the Abstract that should be corrected at this stage (i.e. herpesviruses are not bacteriophages, the portal self-assembles and only requires scaffold to integrate it into the procapsid).

Given the extent of revision needed, we cannot make a decision about publication until we have seen the revised manuscript and your response to the reviewers' comments. Your revised manuscript is likely to be sent for further evaluation by all or a subset of the reviewers.

**IMPORTANT - SUBMITTING YOUR REVISION**

*Re-submission Checklist*

*Published Peer Review*

*PLOS Data Policy*

*Blot and Gel Data Policy*

Sincerely,

Richard

Richard Hodge, PhD

rhodge@plos.org

REVIEWS:

Reviewer #1: PBIOLOGY-D-24-02541R1 

By Co-authors and Hongrong Liu 

The manuscript "Structure of the scaffolding protein and portal within the bacteriophage P22 procapsid provides insights into the mechanism of assembly" by Xiao and co-authors has been submitted to PBIOLOGY. 

Phage P22 is a well recognised model object the structure of which has been studied in different aspects for nearly two decades by many different groups. A role of the scaffolding protein (SP) in bacteriophages is rather well known: it is essential for the self-assembly of phages; but, on the other hand, it is still baffling how does SF initiates that process and what would be a sequence of interactions between SF, portal and capsid proteins. 

The manuscript reports a structure of the SP of phage P22 at a low resolution within the procapsid and describes possible distribution of the SP within the phage procapsid. The authors describe some suggestions on interactions between SP, the portal complex and capsid proteins of the outer shell of the phage. The authors provide some ideas on a possible sequence of self-assemble process related to other phages. 

However it must be mentioned, that the portal protein itself is generally considered to be an initiator of the portal complex, while the scaffolding protein plays a crucial role in facilitating its proper incorporation into a procapsid and guiding the overall assembly process of the viral capsid, rather than initiating the assembly of the portal complex (Dedeo CL, Cingolani G, Teschke CM. Portal Protein: The Orchestrator of Capsid Assembly for the dsDNA Tailed Bacteriophages and Herpesviruses. Annu Rev Virol. 2019 Sep 29;6(1):141-160. doi: 10.1146/annurev-virology-092818-015819). So, the authors of the submitted MS are not right (lines 67-68 and somewhere else), when writing in the abstract (and in the MS), that "the SP catalyses the formation of the dodecameric portal". 

While the topic of this MS is interesting and important, the MS is rather difficult to read, the reasoning of conclusions has to be improved: the authors express certain general ideas, then go to some details, then step back again to a global idea and then return to the details in the figures. Figures should be improved by the proper labelling of details discussed in the MS. 

Several structures were analysed and discussed by the authors; however, they did not submit any atomic models to the EMDB data base which would allow to the reviewers to make their assessments, and to find if any possible confusions can be revealed. These "data" were NOT available.

The methods part is not written in the required style: if one would like to use similar procedures and/or software, the descriptions given by the authors are very sketchy, and the appropriate references are not given. The authors mentioning the references 57, 58, 59, 60, and 61; however, the software links, descriptions of the algorithms, and principles of the assessment of the reconstruction's quality were not provided in these articles. No references were given related to the methods used by the authors of this MS such as Symmetry-mismatch and local reconstruction (the essence of the text in lines 459-477 is extremely far from to be obvious). What was an advantage in usage of these homemade programs and which algorithms that were used in the current analysis was not explained. It seems that the authors were not willing to share their expertise and approaches in image analysis with readers. The lines 483-487 have to be clarified in a more explanatory way: how the local reconstruction has been done, on which algorithms it was based, how many images and what symmetry were used. More details should be given in lines 138-141.

Terminology used in the methods for the data collections and image processing was rather unusual: it would be better if the authors will look at other paper and will used more standard terminology (and for the sample preparation as well).

A set of terms used for the SF domain designation was confusing and badly explained such as a process of oligomerisation, locations of domains in the oligomers: nodule, GOF, column, blade, hook, long loop, helix, base, windmill. It seems that majority of them are overlapping. Please provide the appropriate protein sequences where domains have to be indicated.

It should be written in Methods how the authors identified which band corresponds to the fully assembled phages and which one corresponds to the procapsids (confusing lines 136-137). Info on how these bands were identified should be provided. The representative micrographs must be shown at the same magnification (scale bar must be of the same size!), so a reader will be able to identify which are the procapsids and which ones correspond to the fully assembled phages.

Sizes of particles must be indicated in the text or shown in figures. According to the figures the scale bars given in the images are not correct.

Figures should be improved: partly simplified and better labelled. 

There are many minor comments on this MS. Below is only a part of them:

Line 29. Abstract. Bacteriophages do not include herpesviruses. Rephrase the sentence.

Line 37, The structures shown in the figures were not at "sub-nanometer resolution".

Line 88. What is a homology between SF of the Herpesviruses and P22, where is the pocket located if SP is featureless?

Line 107. "a unique vertex of the coat " -> do the authors have in their mind a mature capsid of P22? P22 does not have any coat compared to the herpes viruses that have lipid coats. 

Line 108, When the ejection proteins are delivered into the phage capsid? How? 

Lines 119-122. This text is related to methods and should be explained there with more details and assessments of criteria used to make the appropriated decisions during refinements.

Lines 156-168. The proper references related to other structures of the P22 connector have to be provided (EMDB numbers and references to the publications), more clear description of the differences between procapsid and mature capsid should be given. Please try to quantify them, indicate differences (shifts and rotations) in the figures.

Line 191-194. Explain in a more transparent way why do you talk of the groups of four (GOF - is a rather confusing term, are you implying on the detective story?) if the SP can make dimers, trimers and why tetrameric oligomers are more important? How in a tetramer of SF (GOF) may have three columns? The arithmetic does not work!

Line 192 "…there are three columns, each of which is located between the two neighboring nodules …? It is extremely confusing, the figure does not help to resolve this confusion.

Line 193 "..From each column, another nodule extends into the external space" (where is this "space"?) Where is "…a stalk-like structure…" , nothing was properly labelled in the figures.

Lines 196-201. The text needs more appropriate language, a schematic figure, and improvement in English. Where are the hook, long loop, and helix? Give the sequences and label these domains in appropriate amino acids as in AA length.

Lines 223-225. Why the trimers interact with the "neighbouring portal subunits"? Do the trimers of the SF are formed only in the vicinity of the portal complex? Why?

Lines 226-241. Does the GOF (four SFs or not?) differ from the windmill? How the structure of the "outer dome" has been obtained? It is confusing: "One blade is a contribution from the side nodule". (??) Figures do not explain much. And now "We speculated that each blade in the windmill-like structure is formed by two base domains, which is similar to the structure of the nodules." It is difficult to follow. One base is supposed to be from one SP? Or the authors had something else in their mind? Should it be that a windmill is equivalent to the GOF? Or it consists of the eight SPs? The authors have to do a better job to explain their ideas.

The sentence on lines 243-247 should be clearer: it is rather confusing in sense of the locations of the SP domains. English must be improved; the explanation should be better.

Lines 259-268. A role and locations of hooks of the SP should be indicated more accurately. The lines are very confusing. Interactions between SF and portal protein have to be somehow justified and better explained.

Discussion should be much more coherent. It raises nearly the same questions as the results part: there is no clear explanations of comparison with other phages and how it can be related to the function of the SP.

Line 292. Why should the SP increase the stability of the portal complex? It is stable even without SFs. 

Line 298 . The nucleation of the capsid is the portal complex, the SP is essential for assembly the icosahedral procapsid.

Lines 303-305. Should be justified and explained in a better way.

Lines 307-311. Should be better explained using results reported in the references.

Conclusions in the discussion should be more clearly formulated and possibly indicated as i, ii, iii and so on. Lines 341-405 do not indicate that some new information was gained during this study compared to the previously published results by the same and other authors in the previous years. 

Of course, it is highly possible that the authors did not describe their achievements in more coherent and well-defined way. 

Reviewer #2: In this manuscript, the authors present the subnanometer resolution structure of the SP and portal in the procapsid of bacteriophage P22. With the AlphaFold2-predicted SP models the authors show that the SP forms trimers and tetramers that assemble into a dome-like structure. Two SP domains are involved in multimerization, with each trimer interacting with two neighboring portal subunits. The SP also features a loop-hook structure that interacts with the coat protein, aiding in its recruitment during assembly. The loops of the SP subunits are positioned in clefts between adjacent portal subunits. The authors suggest that the conformational changes in the portal during maturation likely trigger SP disassembly and release.

The innovation of this manuscript lies in the authors' careful and sophisticated data processing strategies, which enabled the local refinement of SP proteins despite their potentially flexible arrangement. The map and model show strong agreement at subnanometer resolution, and the introduction is clear and well-referenced. However, the presentation of the results could be improved by considering the following suggestions.

Major points:

Related to Figure 1, a supplement movie showing the morphing between the model of the portal in the procapsid and mature virion will be really helpful.

Figure 2 is overly complex, which may make it difficult for readers to interpret. The current color choices could lead to confusion, as the same colors represent different structures across panels. For example, in Panel G, yellow and purple indicate the C-loop and column of gp8, but in Panel I, yellow represents the outer base while purple means the portal. I suggest selecting distinct colors for the gp8 domains in Panel G to avoid this confusion and improve clarity. Additionally, a movie that gradually zooms in and out to highlight the relative positions of the structures in Figure 2 would greatly enhance clarity. As it stands, understanding the spatial relationships and orientations of the different panels is quite challenging from the static figure alone.

The authors mention that it is unclear why only 9 nodules are present on the dodecameric portal. Could it be that the binding of 8 nodules on one side induces a conformational change at the interface, preventing the binding of all the remaining 4 nodules on the other side? If so, could the authors analyze the portal's asymmetry upon SP binding and include a supplementary figure to illustrate this?

Line 246: "approximately 100 models…". Why approximately? How many models are used to generate Fig. S7? 

Minor points:

Line300: what does five hexons mean here since authors provides evidence from literature that coat subunits do not form pentamers and hexamers before assembling into procapsid.

Line 426: To check purity, did the authors run an SDS gel or using negative staining EM?

In Fig. S1B In the image for the upper band of the procapsid, what is the filament-like structure in the micrographs.

S1 Table: symmetry is mispelled and was the dataset for procasid portal used from a previous study (https://www.mdpi.com/1999-4915/15/2/355 supplementary table 1)

Reviewer #3 (Xinghong Dai, signs review): In this article, Hao Xiao et al report the structural studies of phage P22 procapsid, with a focus on the scaffolding protein (SP), and the insights into capsid assembly mechanisms of dsDNA viruses. The successful determination of the SP structure for any spherical phage at secondary-structure resolution represents a great triumph by the symmetry mismatch method, which is pioneered by the senior authors of this paper. The structures are beautifully solved and presented, and the insights brought by these structures are enlightening. I believe this paper would be of great interest to a broad spectrum of readers. 

While this paper is very well written and I enjoyed the reading a lot, here are some suggestions for minor revisions and questions:

1. It would be great if Fig. 1G can be companied with a simple movie morphing from the immature portal to the matured one, preferably showing both the individual subunits and the dodecamers. 

2. The geometry in Fig. 2K is very complicated. It is better to differentiate each subunit of gp8 with different color so that it is easier to understand!

3. Where on the coat protein does the SP hook domain bind? This should be described, preferably with an extra panel in Fig. 4. Is it possible that one SP hook can bind to a coat protein dimer (e.g., if the binding position is across a coat dimer interface)? This seems tempting as 200 SPs vs 415 coat proteins is roughly 1:2. 

4. Is it possible that the position #10 of the SP nodule on the portal, which has very weak densities compared to the others, and the two empty positions, are occupied/hindered by the ejection proteins?

---

## [Decision Letter · Decision Letter 2]

11 Feb 2025

Dear Dr Liu,

Thank you for your patience while we considered your revised manuscript "Structure of the scaffolding protein and portal within the bacteriophage P22 procapsid provides insights into the mechanism of assembly" for publication as a Research Article at PLOS Biology. This revised version of your manuscript has been evaluated by the PLOS Biology editors, the Academic Editor and two of the original reviewers.

Based on the reviews, I am pleased to say that we are likely to accept this manuscript for publication, provided you satisfactorily address the remaining points raised by the Reviewer #1. As indicated in these comments, we would like to strongly encourage you to enlist the services of a professional editing service or a native-speaking colleague to improve the quality of the writing and language in the manuscript text. In addition, please make sure to address the following data and other policy-related requests that I have provided below (A-D):

(A) We would suggest a minor edit to the title, as follows. Please ensure you change both the manuscript file and the online submission system, as they need to match for final acceptance:

“Structure of the scaffolding protein and portal within the bacteriophage P22 procapsid provides insights into the self-assembly process”

(B) Thank you for providing the structural data in the PDB and EMDB databases. However, we note that the data is currently on hold for release. We ask that you please make the structures publicly available at this stage before publication.

(C) Please also ensure that each of the relevant figure legends in your manuscript include information on *WHERE THE UNDERLYING DATA CAN BE FOUND*, and ensure your supplemental data file/s has a legend.

(D) Per journal policy, if you have generated any custom code during the course of this investigation, please make it available without restrictions. Please ensure that the code is sufficiently well documented and reusable, and that your Data Statement in the Editorial Manager submission system accurately describes where your code can be found. 

We expect to receive your revised manuscript within three weeks. 

*Published Peer Review History*

*Press*

Kind regards,

Richard

Richard Hodge, PhD

rhodge@plos.org

Reviewer remarks:

Reviewer #1: Review: PBIOLOGY-D-24-02541R1 

The manuscript (MS) PBIOLOGY-D-24-02541R1 reports on the study of interactions between the scaffolding protein (SP) and the portal complex in phage P22, an assessment of possible distribution of the SP within the phage procapsid using structural analysis by cryoEM. The authors have obtained a few structures at a range of resolutions between 4-8 A. The authors make suggestions which interactions could take place between the SP and the phage portal complex. The topic of this MS is interesting and could be important.

The authors improved the MS, it has now better logistic and is easier to read. However, there are still some issues that make appreciation of the results obtained rather difficult (these issues have been mentioned before and indicated by the second reviewer as well). 

The authors were still not willing to share with readers details related to the image analysis of the asymmetrical elements within the large bio-complexes. Unfortunately, the refences on image processing (given now) do not provide the succinct information as well. How a resolution of the maps obtained was assessed? Where are the FSC curves? How good was the fitting quality of the predicted structures assessed? 

It is interesting, that the authors were concentrated exclusively on interactions of the SP with the portal complex and completely omitting consideration of the SP interactions with the capsid proteins (CP). The SPs induce the CPs oligomerisation, help to form hexameric or pentameric CP oligomers, and then combine them to form the procapsids, this process is accompanied by interactions with other accessory phage proteins. It remains unclear which kind of oligomers form the SPs themselves. It would be useful to have some hypothesis that would explain a mechanism of the self-assembly of procapsids based on the SP activity.

There are many small details that have to be still corrected, mainly of a linguistical sort such as: repeated expressions, strange sentences etc. Somebody has to check the English of the MS and remove unnecessary repeats of words and expressions. The English of some sentences must be corrected.

Reviewer #2: The authors have addressed all my comments.

---

## [Editor Report · Decision Letter 3]

6 Mar 2025

Dear Hongrong,

On behalf of my colleagues and the Academic Editor, David Bhella, I am pleased to say that we can accept your manuscript for publication, provided you address any remaining formatting and reporting issues. These will be detailed in an email you should receive within 2-3 business days from our colleagues in the journal operations team; no action is required from you until then. Please note that we will not be able to formally accept your manuscript and schedule it for publication until you have completed any requested changes.

PRESS

Best regards, 

Richard

Richard Hodge, PhD

rhodge@plos.org

PLOS
